# Phosphatidylcholine Transfer Protein OsPCTP Interacts with Ascorbate Peroxidase OsAPX8 to Regulate Bacterial Blight Resistance in Rice

**DOI:** 10.3390/ijms252111503

**Published:** 2024-10-26

**Authors:** Rong Gong, Huasheng Cao, Yangyang Pan, Wei Liu, Zhidong Wang, Yibo Chen, Hong Li, Lei Zhao, Daoqiang Huang

**Affiliations:** 1Rice Research Institute, Guangdong Academy of Agricultural Sciences, Guangzhou 510640, China; gongrong@gdaas.cn (R.G.); caohuasheng@gdaas.cn (H.C.); yangyangpan@126.com (Y.P.); liuwei@gdaas.cn (W.L.); zhidongwang@163.com (Z.W.); chenyibo@gdaas.cn (Y.C.); gdxxs123@126.com (H.L.); zhaoleijx@163.com (L.Z.); 2Key Laboratory of Genetics and Breeding of High Quality Rice in Southern China (Co-Construction by Ministry and Province), Ministry of Agriculture and Rural Affairs, Guangzhou 510640, China; 3Guangdong Key Laboratory of New Technology in Rice Breeding, Guangzhou 510640, China; 4Guangdong Rice Engineering Laboratory, Guangzhou 510640, China

**Keywords:** START domain, OsPCTP, bacterial blight resistance, reactive oxygen species (ROS)

## Abstract

Rice phosphatidylcholine transfer protein (PCTP), which contains a steroidogenic acute regulatory protein-related lipid transfer (START) domain, responds to bacterial blight disease. Overexpression of *OsPCTP* quantitatively enhances resistance to pathogen in rice, whereas depletion of it has the opposite effect. Further analysis showed that OsPCTP physically interacts with OsAPX8, a ROS scavenging enzyme, and influences ascorbate peroxidase enzymatic activity in vivo. In addition, the expression of pathogenesis-related genes *PR1a*, *PR1b* and *PR10* were significantly induced in *OsPCTP-OE* plants compared with that in wild-type plants ZH11. Taken together, these results suggested that OsPCTP mediates bacterial blight resistance in rice through regulating the ROS defense pathway.

## 1. Introduction

Bacterial blight, caused by the bacterium *Xanthomonas oryzae pv. oryzae* (*Xoo*), is a destructive bacterial disease in rice growth, which can occur at any developmental stage and cause 10% to 50% yield loss depending on the rice variety, growth stage and environmental conditions [1]. Breeding resistant varieties is considered the most economical, effective and sustainable method to prevent and control bacterial blight disease in rice [2]. Generally, plant disease resistance is divided into qualitative (complete) resistance and quantitative (partial) resistance based on the specific interactions of plants against pathogen invasion [3]. Qualitative resistance is conferred by major resistance (*R*) genes with race-specific feature [4]. To date, at least 47 bacterial blight *R* genes have been identified, and 13 of them have been successfully cloned [2,5,6]. However, quantitative resistance belongs to pathogen race-nonspecific resistance and is considered to be more broad-spectrum and durable [7]. The quantitative resistance of rice is the result of the comprehensive effect of various kinds of genes, including antioxidant genes that regulate reactive oxygen species (ROS) metabolism.

When plants are invaded by pathogens, the balance mechanism of ROS metabolism is disrupted by induced oxidative burst. High concentrations of ROS will cause oxidative stress and result in the damage of cellular structures [8]. Then, plants have evolved a series of enzymatic and non-enzymatic antioxidant mechanisms to detoxify in the cell membrane system, thus enhancing its resistance to diseases. These mechanisms of the enzymatic system are mediated by antioxidant enzymes, such as superoxide dismutase (SOD), ascorbate peroxidase (APX), catalase (CAT), glutathione peroxidase (GPX) and peroxyredoxin (PRX) [9]. Previous study showed that *G-BOX BINDING FACTOR 1* (*GBF1*) inhibits the expression of *CAT2*, leading to an increase in H_2_O_2_ content, which induces hypersensitive response and triggers the defense system to pathogens in *Arabidopsis* [10]. The knock-out peroxidase gene *OsPrx30* induces a high concentration of H_2_O_2_ by reducing peroxidase (POD) enzyme activity, resulting in an enhanced ability to defend against bacterial infection [11]. The C_2_H_2_-type transcription factor *Bsr-d1* (Broad-spectrum resistance Digu 1) modulates the rice blast resistance by regulating the transcription of the *Prx* family genes [12]. Overexpression of *OsAPX8* improves the rice pathogen resistance to bacterial blight by enhancing the cell tolerance to H_2_O_2_ [13]. OsAPX1 positively regulates the rice blast resistance through mediating cellular ROS homeostasis with a temporally fine-tuning way. It specifically induces ROS production through affecting the expression of respiratory burst oxidase homologs (*OsRBOHs*) at an early stage of pathogen infection to activate the pathogen defense, while scavenging ROS to eliminate toxicity at a later stage [14]. In addition, WHEAT KINASE START 1 (WKS1) interacts with wheat tAPX to reduce its ability to scavenge active oxygen and results in partial resistance response to stripe rust [15].

Phospholipids are the main components of plant cell membranes, and the unsaturated fatty acids of phospholipids are closely related to plant stress. Phosphatidylcholine transfer protein (PCTP) is a highly specific soluble lipid binding protein that transfers phosphatidylcholines, the highest unsaturated phospholipids class in the membranes, between biological membranes. In animals, PCTP belongs to a steroidogenic acute regulatory protein-related lipid transfer (START) domain family, which has been extensively studied and found to be widely involved in intracellular lipid metabolism, lipid transport and signal transduction processes [16,17]. However, the research is relatively scarce in plants. Previous research showed that there are 35 and 29 START domain proteins in *Arabidopsis* and rice, respectively, which could be divided into three categories based on protein structure [18]. Besides the START domain, Class I proteins also contain the homeodomain (HD) and leucine zipper (Zip) domains, enduing them to be the plant-specific HD-Zip transcription factors that mainly regulate the growth and development of plants. For example, GLABRA 2 (GL2) participates in root epidermal hair growth and development [19,20], PROTODERMAL FACTOR 2 (PDF2) and ARABIDOPSIS THALIANA MERISTEM LAYER 1 (ATML1) are known as mainly regulators for shoot epidermal cell differentiation and embryo development [21,22], and PDF2 also affects floral organ identity [23]. ANTHOCYANINLESS 2 (ANL2) influences epidermal cell development of primary root and anthocyanin accumulation [24]. HOMEODOMAIN GLABROUS (HDG)s, from HDG1 to HDG12, most of them are imprinted proteins and play important effects in plant organs or tissues development, such as floral organ, lateral root, seed, cuticle, shoot meristem and stomata, etc. [23,25,26,27,28,29]. PHAVOLUTA (PHV), PHABULOSA (PHB) and REVOLUTA (REV) play critical roles in lateral meristem initiation and radial positional information in the leaf primordium [30,31]. Rice Class I proteins have conserved functions with their homologous genes in *Arabidopsis,* for example, Roc1 (Rice outermost cell-specific gene 1) and OsTF1 (*Oryza sativa* transcription factor 1) participate in early embryogenesis [32,33], Roc5 controls the leaf rolling by regulating bulliform cell identity [34] and LATERAL FLORET 1 (LF1) regulates the polarity development of lateral organs [35], but OsTF1L (*Oryza sativa* transcription factor 1-like) and Roc4 can modulate drought stress response, which were consistent with the homologous gene AtEDT1/HDG11 in *Arabidopsis* [25,36,37]. The Class II protein ENHANCED DISEASE RESISTANCE 2 (EDR2) with the putative pleckstrin homology (PH) domain was reported to negatively regulate *Arabidopsis* resistance to powdery mildew [38]. However, the biological functions of Subgroup III proteins are currently unclear in plants, which only contain the START domain.

In this study, we identified a Class III protein, OsPCTP, which modulates the bacterial blight resistance of rice. *OsPCTP* positively regulates the pathogen tolerance, and its expression is significantly induced by *Xoo.* Further study found that OsPCTP affects the activity of the APX enzyme by interacting with OsAPX8, thus influencing the content of H_2_O_2_ in rice. In conclusion, our study revealed the regulatory function of the START domain protein in rice bacterial pathogen interactions.

## 2. Results

### 2.1. Identification and Characterization Analysis of OsPCTP

PCTP, also called STARD2 (START-domain-containing 2), belongs to the START domain superfamily, which transfers phosphatidylchloine by specific lipid ligands between membranes and is widely researched in animals [17,39], whereas few studies have been conducted in plants. Therefore, we isolated *OsPCTP* from the rice genome based on the sequence of Loc_Os02g26860 (Appendix A), which was obtained through applying a BLASTP query with human PCTP protein sequence in the database (NCBI, https://www.ncbi.nlm.nih.gov/ accessed on 12 March 2021). The *OsPCTP* coding sequence (CDS) is 1308 bp in length, including six exons and five introns (Appendix A), and its encoded protein has 435 amino acids with an estimated molecular weight of 48.38 kDa. Protein structure prediction indicates that OsPCTP also contains a transmembrane domain in the N-terminus in addition to the START domain, compared with human PCTP (Appendix A). To explore the candidate START domain family in rice, we queried the structural domain number PF01852 in the Rice Genome Annotation Project website (RGAP) (https://rice.plantbiology.msu.edu/ accessed on 12 March 2021) and ultimately identified 26 correlative proteins. The 15 protein sequences of the START domain family from humans were downloaded from NCBI. To elucidate the phylogenetic relationship between OsPCTP and other START domain genes, a neighbor-joining phylogenetic tree was constructed of these proteins based on the p-distance model and pairwise deletion algorithm. The result showed that these proteins are divided into four groups. OsPCTP and LOC_Os07g08760 share the highest degree of homology with human STARD proteins, especially with STARD2-like proteins (Figure 1).

The START domain proteins from rice were classified into five subgroups according to their structures and sizes. The majority of START domain proteins are simultaneously classified in the HD-ZIP transcription factor family due to containing the additional homeodomain and leucine zipper (bZip or ZLZ) domain, and they are further divided into HD-ZIP III (eight members) and HD-ZIP IV (twelve members) subcategories according to the ZIP structural difference. The other small subclass is PH-START (three members), which contains PH and EDR2_C (Protein EDR2, C-terminal) domains in addition to the START domain. Interestingly, only three proteins just contain the START domain. OsPCTP is one of them, which has the closest phylogenetic relationship to the PH-START subclass (Figure 2). Orthologous genes *EDR2* and *WKS1* were reported to modulate pathogen resistance in *Arabidopsis* and wheat, respectively [15,38]. Hence, we speculated that OsPCTP may play a role in plant innate immunity.

### 2.2. Expression Patterns of OsPCTP

Previous studies have shown that the HD-START genes play crucial roles in rice growth and development [32,33,34,35,36,37]. To obtain insight into the potential function of the *OsPCTP*, we performed quantitative real-time PCR (qRT-PCR) to examine the expression profile of *OsPCTP* in various rice tissues, including the seedling and root from 14 d old, leaf, leaf sheath, stem and inflorescence from 65 d old rice plants. The qRT-PCR data showed that *OsPCTP* was expressed in all tested tissues, with the highest expression in seedling and leaf and with the lowest expression in root (Figure 3A). Lipid transport genes play an important role in the stress response process of higher plants, and their expression could be induced by environmental factors [40,41,42]. Therefore, we downloaded the RNA-seq data of rice seedlings with stress and phytohormomes treatment from the RAPDB database (https://rapdb.dna.affrc.go.jp/ accessed on 18 May 2021) to generate the heatmap. As shown in Appendix A, the expression of *OsPCTP* is remarkably induced by environmental stress, such as cold, osmosis, high salinity and flood. In addition, we found it is also induced by jasmonic acid (JA) and abscisic acid (ABA). We further analyzed the promoter region of the *OsPCTP* and found it has three ABA responsive and one salicylic acid (SA) responsive elements, respectively (Appendix A). JA and SA are the major defense signaling compounds mediating disease resistance in plants [43,44]. To investigate whether *OsPCTP* is involved in bacterial blight resistance, the expression of *OsPCTP* was detected at 0, 6, 24, 48 and 72 h after bacterial blight inoculation. The result showed that the transcription level of *OsPCTP* was first suppressed at 6 h, then rapidly and significantly induced until 48 h, reaching its peak, and then gradually decreased after infection at 72 h (Figure 3B). These observations suggested that OsPCTP is involved in bacterial blight resistance.

### 2.3. Overexpression of OsPCTP Improves Resistance to Bacterial Blight

To elucidate the possible contribution of OsPCTP in rice immunity against bacterial blight, we first investigated whether the alteration of *OsPCTP* expression would cause changes in bacterial blight resistance. Both *OsPCTP* knockout and overexpression (*OsPCTP-OE*) transgenic rice plants were constructed with the variety ZH11. Genotype detection indicated that *OsPCTP* knockout plants (*ospctp*) are the homozygous deletion mutant with a C/G deletion in the second exon, resulting in the premature stop codon (Figure 4A). The qRT-PCR result showed that the expression of *OsPCTP* was prominently elevated in the two independent *OsPCTP-OE* transgenic lines (OE1 and OE2) (Figure 4C). After inoculation with the Chinese *Xoo* race 4 of rice blight, *ospctp* mutants were susceptible to rice blight with larger lesion length, whereas OE1 and OE2 lines showed a significantly increased resistance compared with the ZH11 plants (Figure 4B,D). The results indicated that OsPCTP acts as a positive regulator of quantitative immunity in the bacterial blight resistance of rice.

### 2.4. OsPCTP Interacts with Ascorbate Peroxidase OsAPX8

To further elucidate the molecular mechanism of OsPCTP in bacterial blight resistance, we explored OsPCTP-interacting proteins by screening a yeast library and obtained a protein OsAPX8, which is an ascorbate peroxidase. Previous study found that OsAPX8 participates in the biological regulation process of the resistance to bacterial pathogen and salt osmotic stress [13]. Therefore, we presumed that OsPCTP regulated bacterial blight tolerance, possibly through interacting with OsAPX8. To verify this, we first constructed the yeast expression vectors pGADT7-OsAPX8 and pGBDT7-OsPCTP for a yeast two-hybrid assay. The assays showed that BD-OsPCTP could interact with AD-OsAPX8 but not with AD or BD empty carrier (Figure 5A). The interaction was further confirmed in vivo by a co-immunoprecipitation (co-IP) assay. Tobacco (*Nicotiana benthamiana*) leaves expressing OsPCTP-GFP/OsAPX8-FLAG and GFP/OsAPX8-FLAG were harvested to isolate total proteins followed by an immunoprecipitation assay, respectively. Western blotting analysis showed that OsAPX8-FLAG could be successfully co-precipitated with OsPCTP-GFP, but not with GFP tag by GFP antibody (Figure 5B). In addition, this interaction was also corroborated by firefly luciferase complementation imaging (LCI) assay. As shown in Figure 5C, the co-expression of cLUC-APX8 with PCTP-nLUC generated strong LUC signals in tobacco leaves, whereas no obvious signal was observed from the co-expression of the negative controls in which cLUC-APX8 was co-expressed with nLUC or PCTP-nLUC was co-expressed with cLUC, consistent with the quantitative analysis of luciferase activity (Figure 5D). The result further supported the association of OsPCTP and OsAPX8 in plant cells.

### 2.5. OsPCTP Influences the Enzyme Activity of APX

To dissect the potential role of the OsPCTP–OsAPX8 interaction, we firstly examined the *OsAPX8* expression levels in *OsPCTP* transgenic and ZH11 plants at different time points (0, 6, 24, 48 and 72h) after *Xoo* infection. The results showed that *OsAPX8* expression initially was suppressed and then rapidly induced after 6 h until 48 h, then gradually decreased to the original state at 72 h, consistent with the expression trend of *OsPCTP* after inoculation (Figure 3B). However, there was no dominant difference between the transgenic plants and control at all time points (Figure 6A). To further detect whether OsPCTP affects the APX enzyme activity, we analyzed the APX enzyme activity in *OsPCTP* transgenic and control plants before and after blight infection. The results indicated that APX enzyme activity was significantly enhanced in OsPCTP-OE lines compared with control ZH11 both before and after infection. However, these was no obvious difference between *ospctp* and ZH11 (Figure 6B). Thus, we deduced that OsPCTP interacts with OsAPX8 to affect APX enzyme activity, thereby participating in its mediated bacterial blight resistance pathway.

### 2.6. OsPCTP Influences H_2_O_2_ Content and Modulates the Expression of PR Genes

The ascorbate–glutathione cycle mediates hydrogen peroxide (H_2_O_2_) detoxification in chloroplasts and cytosol, of which APX is the key enzyme and encoded by *APXs* genes [45]. APXs can catalyze the conversion of H_2_O_2_ to H_2_O and O_2_ using ascorbate as the specific electron donor [46]. To investigate whether OsPCTP influences the accumulation of ROS, we examined the content of H_2_O_2_ in transgenic materials before and after bacterial blight inoculation. As shown in Figure 7A, H_2_O_2_ content exhibited prominent reduction in the *OsPCTP-OE* lines compared with control plants ZH11, both before and after infection, whereas there was no significant difference between *ospctp* and ZH11. As the most stable ROS, H_2_O_2_ also serves as a regulatory factor for the disease defense-related genes in signal transduction. To further investigate whether *OsPCTP* participates in the modulation of defense-related genes, we detected the expression of three pathogenesis-related (*PR*) genes (*PR1a*, *PR1b* and *PR10*) and the SA synthesis gene phenylalanie ammonia-lyase (*PAL1*) in the *OsPCTP* transgenic plants before and after blight infection at 6 h and 24 h. *PR1a*, *PR1b* and *PR10* are the well-characterized *PR* genes in rice. The expression of *PR1* genes is often used as a marker for plant systemic acquired resistance (SAR) development [47,48]. *PR10* encodes ribonuclease, which is considered to be able to degrade viral RNA during the plant disease resistance process [49]. The result showed that these defense-related genes were rapidly induced after blight infection, and their expression levels were significantly upregulated in the *OsPCTP-OE* lines and downregulated in the *ospctp* mutant compared with the control plants ZH11. In addition, the expression of *PR1a*, *PR1b* and *PR10* were also prominently higher in *OsPCTP-OE* lines than ZH11 before infection, while there were no obvious difference of *PAL1* between transgenic plants and ZH11 (Figure 7B). Thus, we guessed that OsPCTP probably involved in ROS metabolism through OsAPX8, thereby regulating defense-related genes in disease resistance responses.

## 3. Discussion

Among the 26 START domain proteins of rice, 20 of them contain homeodomain and leucine zipper domains excepting the START domain, belonging to the HD-ZIP transcription factors at same time (Figure 2). These homologous proteins perform various regulatory activities in multiple aspects of rice growth and development, such as the epidermis differentiation, lateral organ polarity, leaf initiation process, lignin biosynthesis and stomatal closure [32,33,34,35,36,37]. However, OsPCTP only contains the START domain, and its function has not been reported in rice. In current study, we found that *OsPCTP* expression is induced by bacterial blight disease (Figure 3B), as well as abiotic stress such as cold, salt, osmosis and also plant hormones induction like ABA and JA (Appendix A). In addition, we found that ABA, SA and low temperature responsive elements are included in the promoter region of *OsPCTP* (Appendix A). Hence, we speculated that OsPCTP may be involved in various stress conditions in rice.

To explore the function of OsPCTP in biotic stress, we constructed transgenic plants of *OsPCTP-OE* and *ospctp* mutants and found overexpression of *OsPCTP* significantly improved bacterial blight resistance, whereas mutants would enhance susceptibility. *OsPCTP-OE* lines displayed shorter lesion length during bacterial blight infection compared with control plants ZH11 and *ospctp* mutants (Figure 4). Thus, we concluded that *OsPCTP* is a positive regulator of rice bacterial blight resistance. To further elucidate the disease resistance mechanism of OsPCTP, we obtained an interacted protein OsAPX8 by yeast library screening and then validated the interaction through yeast two-hybrid assay, luciferase complementation assay and co-immunoprecipitation experiment. OsAPX8 is the key enzyme in the ROS detoxification system, and overexpression of *OsAPX8* enhances tolerance to bacterial blight and high salt stress in rice [13]. Furthermore, the expression pattern of *OsAPX8* is consistent with *OsPCTP* during pathogen infection (Figure 3B and Figure 6A). These findings support the hypothesis that OsPCTP mediates bacterial blight resistance through interacting with OsAPX8. To further dissect the functional connection between OsPCTP and OsAPX8 in the regulation of bacterial pathogen resistance, we detected the expression lee of a transcription factor in el of *OsAPX8* and APX enzyme activity. The results indicated that *OsPCTP* did not affect the expression of *OsAPX8* but influenced the APX enzyme activity. APX enzyme activity significantly increased in *OsPCTP-OE* lines compared with control plants ZH11 both before and after bacterial infection (Figure 6A,B). As a major antioxidant enzyme, APX plays an important role in the H_2_O_2_-scavenging and ROS dynamic balance [45,46]. Most researches showed that APXs can reduce excessive intracellular ROS triggered by kinds of biotic or abiotic stresses, thereby enhancing tolerance of rice to stress [13,14,50,51]. In this study, we found that H_2_O_2_ contents in leaves were prominently lower in the *OsPCTP-OE* lines than in control ZH11 before and after infection (Figure 7A), consistent with the trend of APX enzyme activity (Figure 6B). It has been recognized that H_2_O_2_ was the most important factor for the serious leaf dehydration and withering of rice without major resistance genes and was not the cause of hypersensitivity [13]. Taken together, our results demonstrated that OsPCTP mediated pathogen quantitative resistance by interacting with OsAPX8 and influencing the APX enzyme activity, as to modulate ROS homeostasis in rice.

Surprisingly, *ospctp* mutants displayed more susceptibility to pathogen infection than control plants (Figure 4D), although there are no significant differences in APX enzyme activity and H_2_O_2_ content between them (Figure 6B and Figure 7A). We speculated that there could be other *APXs* genes which complement the effects produced by *OsPCTP* knockout, which needs to be further researched. In addition, we discovered that the expression levels of the defense-related genes, including *PAL1*, *PR1a*, *PR1b* and *PR10,* which are also involved in the SA-dependent signaling pathway [43], were significantly lower in *ospctp* mutants than in ZH11 plants (Figure 7B). Previous research demonstrated SA is an archetypal defense hormone and its importance in the plant innate immunity is well documented [43,52]. Moreover, we discovered the SA and ABA response elements in the promoter region of the *OsPCTP*. Recent studies showed that ABA also plays an important role in plant biotic stress, excepting abiotic stress [53,54,55]. In general, we conjectured that OsPCTP is primarily involved in basal defense through affecting the ROS homeostasis in cell, but there might be other mechanism that cooperatively regulates the disease resistance of *OsPCTP* at same time, which needs to be further verified.

## 4. Materials and Methods

### 4.1. Plant Materials and Growth Conditions

*OsPCTP-OE* lines and *ospctp* knockout lines were generated in japonica cultivar ZH11. All plants were grown at the Rice Research Institute, Guangdong Academy of Agricultural Sciences (23.13° N, 113.27° E), Guangzhou, China. The blight bacterium Chinese *Xoo* race 4 isolate collected from Guangdong, China, was used for inoculation. To evaluate bacterial blight disease, plants were inoculated with Chinese *Xoo* race 4 (OD600 = 0.3) at the tillering stage by the leaf-clipping method [56]. Disease was scored by measuring the length of disease lesions on the leaves at 12 d after inoculation.

### 4.2. Data Search and Phylogenetic Analysis

To obtain the protein sequencing data for all of the START domain genes in rice, the structural domain number PF01852 was queried in the Rice Genome Annotation Project website (RGAP) (http://rice.plantbiology.msu.edu/ accessed on 12 March 2021). The protein sequences of START domain family genes from humans were downloaded from NCBI (http://ncbi.nlm.nih.gov/ accessed on 12 March 2021). Phylogenetic analysis was performed using MEGA 11 by the neighbor-joining algorithm with 1000 bootstrap replicates based on amino acid sequences. The analysis of protein structure was conducted in SMART (http://smart.embl-heidelberg.de/ accessed on 12 March 2021) and InterPro (https://www.ebi.ac.uk/interpro/search/sequence/ accessed on 12 March 2021).

### 4.3. RNA Extraction and qRT-PCR Analysis

Total RNA was extracted from the collected samples using the Hipure plant RNA mini Kit (Magen, Guangzhou, China) according to the manufacturer’s instructions. Then, the first-strand cDNA was reverse-transcribed with 1 μg of total RNA using PrimeScript RT reagent Kit (Transgen, Beijing, China) based on the manufacturer’s instructions. qRT-PCR was performed in a 96-well or 384-well plate using SYBR premix ExTaqTM (Takara Bio, Beijing, China) on a CFX Connect real-time PCR detection system (Bio-Rad, San Francisco, CA, USA). Relative expression of the target genes was calculated using the 2^−∆∆CT^ method. Each experiment was carried out with three biological repeats and rice Ubiquitin (*UBQ*) was used as an internal control. Primers used for qRT-PCR are listed in Appendix A.

### 4.4. Yeast Two-Hybrid Assay

The yeast two-hybrid (Y2H) assay was performed following the manufacturer’s instructions (Clontech, Shanghai, China). To generate AD-OsAPX8, the full-length CDS of OsAPX8 was inserted into the pGADT7 vector through EcoR I/Xho I restriction sites. For BD-OsPCTP, the full-length CDS of OsPCTP was amplified and inserted into the pGBKT7 vector by the same restriction sites. The resulting constructs were co-transformed into the yeast strain AH109. The plasmid combination of pGBKT7-53 and pGADT7-T was used as a positive control. All yeast transformants were grown on SD/-Trp/-Leu medium and screened on SD/-Trp/-Leu/-His/-Ade medium for the interaction test at 30 °C for 3 d. Primers used for vector construction are listed in Appendix A.

### 4.5. Luciferase Complementation Imaging Assays

The coding sequences of *OsPCTP* and *OsAPX8* were separately cloned into the JW771 (PCAMBIA2300-nLUC) and JW772 (PCAMBIA2300-cLUC) vectors by the Kpn I and Sal I restriction sites to generate OsPCTP-nLUC and cLUC-OsAPX8 fused vectors, respectively. These plasmids were transformed into Agrobacterium tumefaciens GV3101. A. tumefaciens containing OsPCTP-nLUC, A. tumefaciens containing cLUC-OsAPX8 and A. tumefaciens containing P19 were co-injected into fully expanded leaves of 6-week-old N. Benthamian plants at a ratio of 2:2:1, and the plasmid combination of AtFLS2-nLUC and AtAGB1-cLUC was used as the positive control. After infiltration for 48 h, the infiltrated regions were injected with 1 mM of luciferin (Promega, Beijing, China). The luminescence activity was captured with a chemiluminescent imaging system (Tanon, Shanghai, China). Primers used for vector construction are listed in Appendix A.

### 4.6. Co-Immunoprecipitation

The full-length coding sequence of *OsPCTP* was inserted into the PCAMBIA1300-GFP vector, while the coding sequence of *OsAPX8* was inserted into the P1306-Flag vector to express OsPCTP-GFP and OsAPX8-FLAG, respectively. The recombinant plasmids were transformed into Agrobacterium tumefaciens by strain GV3101 and then infiltrated into 5-week-old N. benthamiana leaves. The total proteins were extracted from leaves expressing OsPCTP-GFP/OsAPX8-FLAG or GFP/OsAPX8-FLAG constructs with protein extraction buffer [50 mM Tris-HCl (pH 7.5), 150 mM NaCl, 1 mM EDTA, 10% glycerol and 1% NP-40] containing 1×protease inhibitor cocktail. An anti-GFP antibody was added to the protein extracts for immunoprecipitation for 3 h, then protein A magnetic beads (36403ES03, Yeasen, Shanghai, China) were added and incubated for another 2 h. After washing three times with the protein extraction buffer, the co-immunoprecipitated products were separated by SDS-PAGE gel and detected with anti-GFP (HT801,TransGen, Beijing, China; 1:5000) and anti-FLAG (A2220, Sigma, Shanghai, China; 1:5000) antibodies. Primers used for vector construction are listed in Appendix A.

### 4.7. Analysis of Enzyme Activity and Measurement of H_2_O_2_ Content

The APX enzyme activity was examined with an assay kit (Sangon Biotech, Shanghai, China) following with manufacturer’s instructions. Fresh leaves with a length of 3 cm including the inoculation wound at the rice tillering stage were collected and ground with liquid nitrogen. The 1 mL lysis buffer solution was added to 100 mg of leaf powder and blended fully. The supernatant was collected by centrifuging at 13,000× *g* for 20 min at 4 °C. Then, the 20 μL of supernatant and 180 μL of test solution were transferred to the test-tubes immediately, and the mixed solution was used to measure the oxidation rate of ascorbic acid within 2 min with a spectrophotometer at 290 nm using a Thermo Scientific Multiskan Spectrum (Thermo, Waltham, MA, USA).

H_2_O_2_ was measured using the hydrogen peroxide assay kit (Grace Biotechnology, Suzhou, China) according to the manufacturer’s instruction. Briefly, 100 mg of fresh leaf tissue as described above was ground fully in 1 mL cold acetone and centrifuged to remove cellular debris. The 250 μL of supernatant was extracted with 75 μL and 50 μL Reagent I and Reagent II at room temperature. Then, the extract was fully dissolved with 230 μL Reagent III, and the absorbance was measured with 200 μL of solution by measuring the absorbance at 415 nm using a Thermo Scientific Multiskan Spectrum (Thermo, USA).

## 5. Conclusions

In the present study, we elucidated the biological function of the START domain protein, OsPCTP, in bacterial blight resistance of rice. OsPCTP functions as a positive regulator to modulate disease tolerance in a quantitative manner, and the OsPCTP-mediated pathogen resistance is associated with the regulation of ROS homeostasis. OsPCTP interacts with OsAPX8, a ROS scavenging enzyme, to affect the APX enzyme activity, thereby mediating the H_2_O_2_ metabolism in rice. Interestingly, the *ospctp* mutant plants showed reduced resistance to bacterial blight, but there were no significant differences in APX enzyme activity and H_2_O_2_ content compared with the control ZH11. Moreover, the expression levels of disease defense-related genes in the SA-dependent signaling pathway were significantly decreased in *ospctp* mutants. Further analysis of the *OsPCTP* promoter revealed that it contains the regulatory elements of SA and ABA, which play important roles in plant biotic and abiotic stress. Overall, we speculated that OsPCTP may mediate rice resistance to bacterial blight through multiple pathways in a collaboratively regulating manner, which requires further validation.

## Figures and Tables

**Figure 1 ijms-25-11503-f001:**
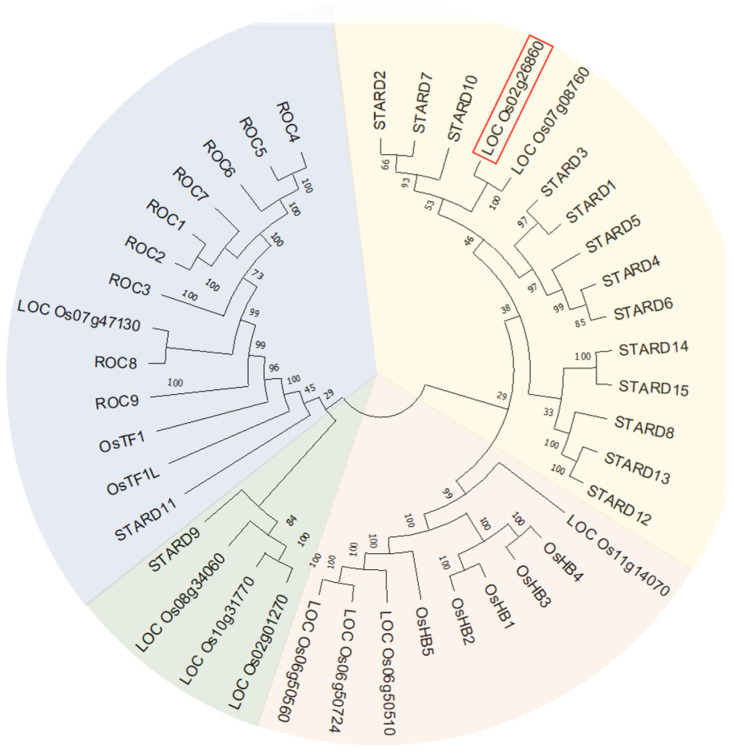
Phylogenetic analysis of the START domain containing proteins between rice and humans. A neighbor-joining phylogenetic tree was constructed based on the p-distance model and pairwise deletion algorithm (bootstrapped 1000 replicates). The locus name of OsPCTP is marked with red box.

**Figure 2 ijms-25-11503-f002:**
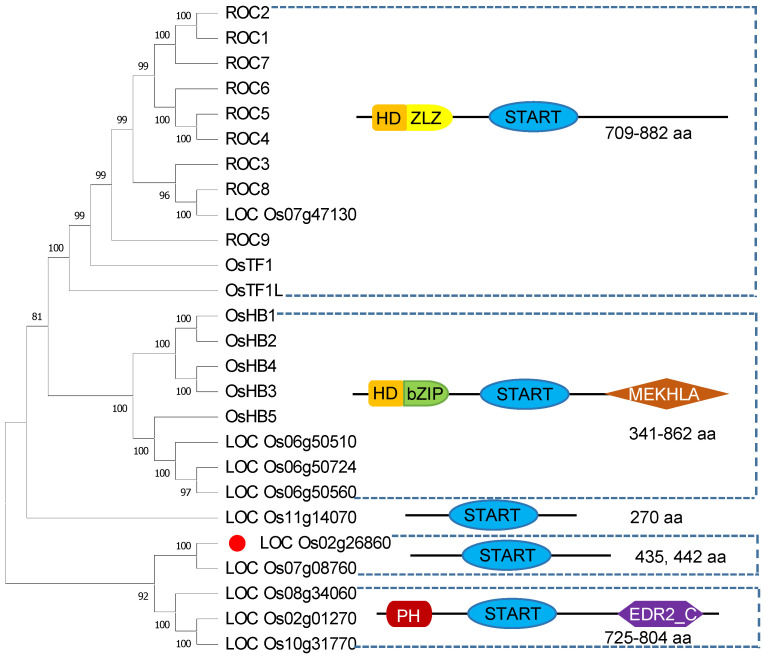
START domains from 26 rice START-containing proteins are divided into five subfamilies. The structure and domain for each protein subfamily is shown on the right. Sizes of the corresponding proteins in amino acids (aa) are indicated to the right or below each representational. The locus name of OsPCTP is marked with red box.

**Figure 3 ijms-25-11503-f003:**
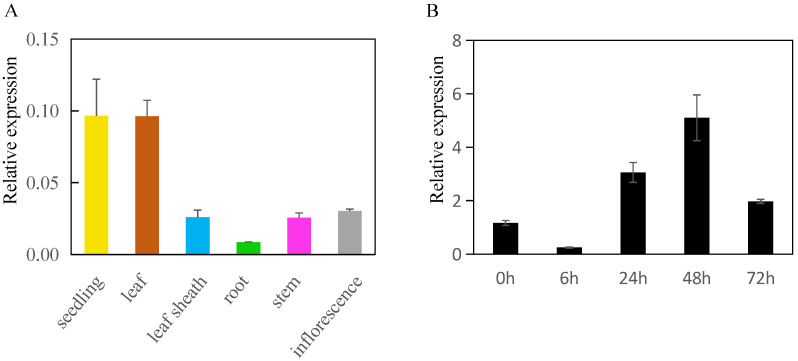
Expression patterns of *OsPCTP*. (**A**) Expression pattern of *OsPCTP* in different tissues of ZH11 plants. Seedling and root at 14 d, other tissues at 65 d after sowing. (**B**) Quantitative RT-PCR analysis of the response of *OsPCTP* to bacterial blight infection. At the tillering stage, about 3 cm length leaves under the inoculation wound were used for gene expression analysis. The values 6 h, 24 h, 48 h and 72 h indicate the time after bacterial blight inoculation. Error bars indicate the SD from three biological replicates.

**Figure 4 ijms-25-11503-f004:**
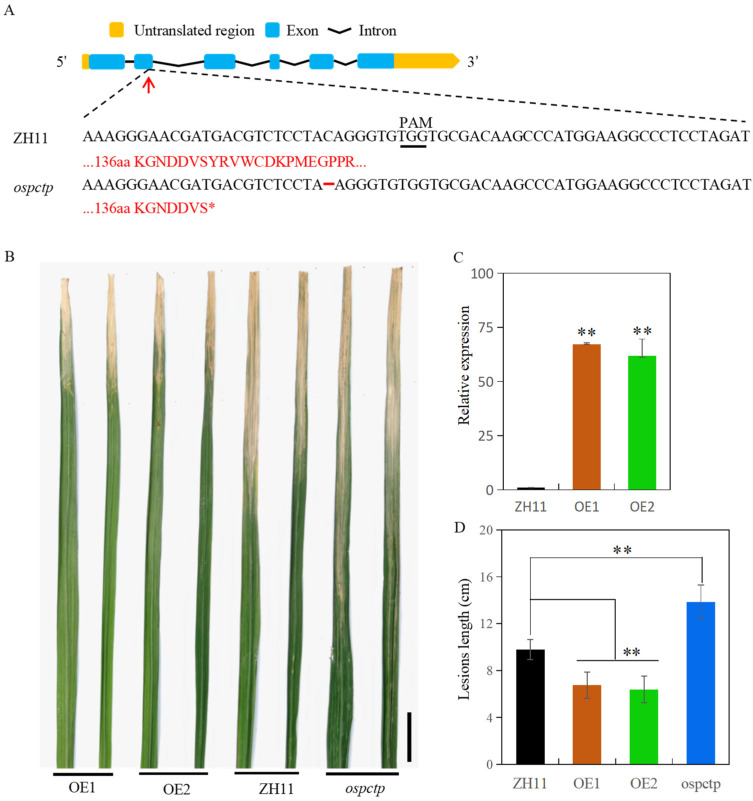
*OsPCTP* positively regulates rice bacterial blight resistance. (**A**) Schematic of the *OsPCTP* CRISPR/Cas9 mutants. The red short line indicates base deletion. Red letters indicate encoded amino acids. Asterisks indicates stop codon. The black underline indicates the PAM. The red arrow reprsents the target of gene editing. PAM, protospacer adjacent motif. (**B**) Phenotypes of *OsPCTP* transgenic and ZH11 plant at 12 d after inoculation with *Xoo* inoculation at the tillering stage. Scale bar: 2 cm. (**C**) qRT-PCR analysis of *OsPCTP* expression in ZH11 and two *OsPCTP-OE* lines. Rice leaves in the tillering stage were used for gene expression analysis. Ubiquitin was used as an internal control. The data are the mean ± SD (n = 3). (**D**) Lesions length in the *OsPCTP* transgenic and control plants. Error bars indicate the SD from ten biological replicates. ** *p* < 0.01.

**Figure 5 ijms-25-11503-f005:**
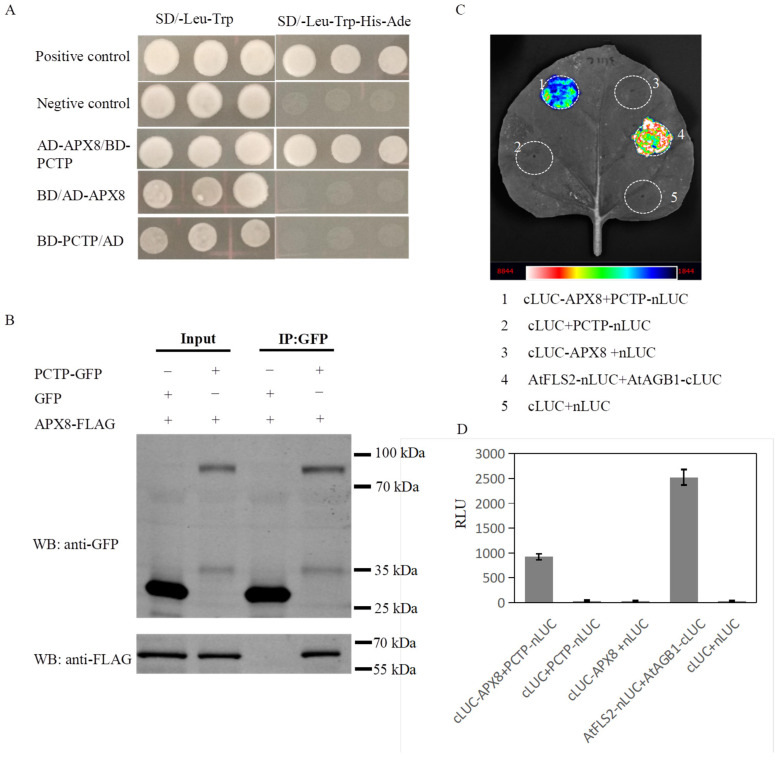
OsPCTP physically interacts with OsAPX8. (**A**) Yeast two-hybrid analysis the interaction of OsPCTP and OsAPX8. (**B**) Co-immunoprecipitation assay reveals that OsPCTP associates with OsAPX8 in vivo. (**C**,**D**) The LCI assay shows that OsPCTP interacts with OsAPX8 in Nicotiana benthamiana leaves by transient expression. Error bars indicate the SD from three biological replicates.

**Figure 6 ijms-25-11503-f006:**
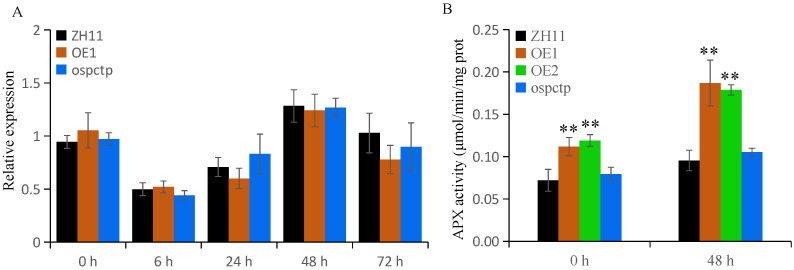
OsPCTP increases APX enzyme activity but does not regulate the expression level of *OsAPX8*. (**A**) Expression level of *OsAPX8* was detected in the ZH11 and *OsPCTP* transgenic plants at 72 h after *Xoo* inoculation. Error bars indicate the SD from three biological replicates. (**B**) APX enzyme activity in *OsPCTP* transgenic and ZH11 plants. Values are means ± SD of three independent experiments; asterisks represent a significant difference compared with the control plant. ** *p* < 0.01.

**Figure 7 ijms-25-11503-f007:**
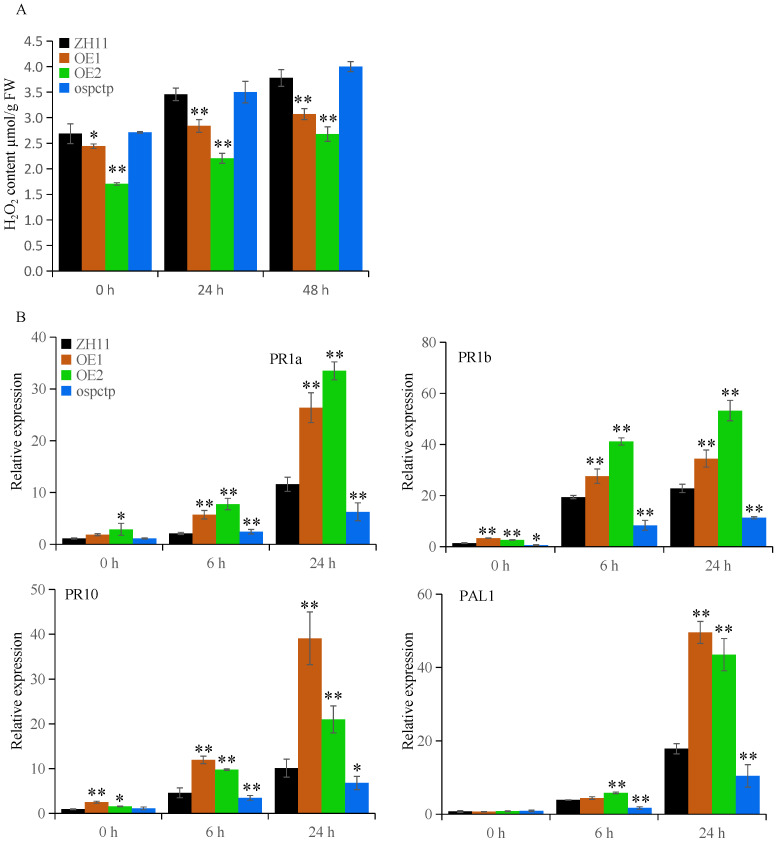
OsPCTP influences H_2_O_2_ content and modulates the expression of PR genes. (**A**) H_2_O_2_ content assay in the *OsPCTP* transgenic and ZH11 plants. Values are means ± SD of three independent experiments; asterisks represent a significant difference compared with the control plants (* *p* < 0.05, ** *p* < 0.01). (**B**) Relative expression analysis of *PR* genes (*PR1a*, *PR1b* and *PR10*) and *PAL1* in ZH11 and *OsPCTP* transgenic plants. Error bars indicate the SD from three biological replicates (* *p* < 0.05, ** *p* < 0.01).

## Data Availability

Data are contained within article and Appendix A.

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
