# Peer review of "Phosphatidylcholine Transfer Protein OsPCTP Interacts with Ascorbate Peroxidase OsAPX8 to Regulate Bacterial Blight Resistance in Rice"

_ijms, 2024, doi:10.3390/ijms252111503_

Round 1
Reviewer 1 Report
Comments and Suggestions for Authors
The work is convincing enough, but many details are absent, and a number of revisions are needed.
- The first 4 lines of the abstract should be deleted/edited by starting with something like “The study of rice Phosphatidylcholine transfer protein, which contains an acute regulatory protein-related lipid transfer (START) domain, responds to bacterial blight disease ..... ” avoiding overemphasis on the START domain.
- Acronyms could be reduced in the title, for example, with “Phosphatidylcholine Transfer Protein (PCTP) Interacts with Ascorbate Peroxidase to Regulate Bacterial Blight Resistance in Rice”.
- The meaning of all acronyms should be explained at the first insertion (e.g. GBF1, Brs-d1, WKS, GL2, PDF2, ATML, ANL2, HDGs, STARD, …..)
- Figure 1 legend: it is written that “OsPCTP was marked out with red texbox”, but within the figure it is indicated as LOC_Os02g26860.
- Righe 156-157: it is not indicated the tissue and the timing.
- Figure 3A: it is necessary to indicate the age of the tissues. Also, please use a different color for each tissue.
- Figure 3B: it is necessary to indicate the tissue and eventually the leaf portion (or if a whole leaf was analyzed).
- Figure 4 legend: it is necessary to indicate the age / number of the leaves and the time after inoculation for B, C and D. In addition, please use a different histogram color for each transgenic line.
- Figure 5D is of low quality, please revise.
- Figure 6: it is opportune to use histograms for Fig. 6A and use different colors for 6A and 6B in accordance with Fig. 4.
- Figure 7: please use colored histograms. For 7A use the symbol “µ” for the H2O2 content unit.
- Lines 391-393 and 399: It is difficult to understand how 100 mg leaf was weighed after homogenization with liquid nitrogen. In addition, 100 mg FW of leaf corresponds to a portion of a leaf, certainly not a whole leaf. It should be better explained and indicated which leaf was taken and at what age of the plant. “100 mg fresh leaf tissue” is too general.
Author Response
Comments 1:The first 4 lines of the abstract should be deleted/edited by starting with something like “The study of rice Phosphatidylcholine transfer protein, which contains an acute regulatory protein-related lipid transfer (START) domain, responds to bacterial blight disease ..... ” avoiding overemphasis on the START domain.
Response 1: Thank you for pointing this out. We have focused on the role of lipids in plant growth, development, and stress response for a long time. The phosphatidylcholine transfer protein (PCTP) in animals belongs to the START-domain containing family and has a typical START domain. This domains are characterized as a hydrophobic ligand binding pocket and are involved in lipid/sterol binding, transport, and signaling in both animal and plant species. Previous studies showed that the START domain proteins have important biological functions in rice, especially in plant growth and development. However, the functions of disease resistance for START domain proteins have not been reported yet. In addition, we have confirmed OsPCTP could bind to phosphatidylcholine in vitro experiments (data not shown), which is the typical feature of the START domain protein. Therefore, we adopted the above description in abstract for providing a detailed introduction of the START domain.
Comments 2: Acronyms could be reduced in the title, for example, with “Phosphatidylcholine Transfer Protein (PCTP) Interacts with Ascorbate Peroxidase to Regulate Bacterial Blight Resistance in Rice”.
Response 2: Thank you for pointing this out. We have made correction according the reviewer’s suggestion.
Comments 3: The meaning of all acronyms should be explained at the first insertion (e.g. GBF1, Brs-d1, WKS, GL2, PDF2, ATML, ANL2, HDGs, STARD, …..)
Response 3: Thank you for pointing this out. We have made correction according the reviewer’s suggestion.
Comments 4: Figure 1 legend: it is written that “OsPCTP was marked out with red texbox”, but within the figure it is indicated as LOC_Os02g26860.
Response 4: Thank you for pointing this out. We have made correction according the reviewer’s suggestion.
Comments 5: Righe 156-157: it is not indicated the tissue and the timing.
Response 5: Thank you for pointing this out. We have made corresponding supplements in manuscript according the reviewer’s suggestion.
Comments 6: Figure 3A: it is necessary to indicate the age of the tissues. Also, please use a different color for each tissue.
Response 6: Thank you for pointing this out. We have made correction and supplements in figure 3A according the reviewer’s suggestion.
Comments 7: Figure 3B: it is necessary to indicate the tissue and eventually the leaf portion (or if a whole leaf was analyzed).
Response 7: Thank you for pointing this out. We have made supplements in figure 3B according the reviewer’s suggestion.
Comments 8: Figure 4 legend: it is necessary to indicate the age / number of the leaves and the time after inoculation for B, C and D. In addition, please use a different histogram color for each transgenic line.
Response 8: Thank you for pointing this out. We have made correction and supplements in figure 4 following the reviewer’s suggestion.
Comments 9: Figure 5D is of low quality, please revise.
Response 9: Thank you for pointing this out. We have made correction in figure 5B according the reviewer’s suggestion.
Comments 10: Figure 6: it is opportune to use histograms for Fig. 6A and use different colors for 6A and 6B in accordance with Fig. 4.
Response 10: Thank you for pointing this out. We have made correction in figure 6 according the reviewer’s suggestion.
Comments 11: Figure 7: please use colored histograms. For 7A use the symbol “µ” for the H2O2 content unit.
Response 11: Thank you for pointing this out. We have made correction in figure 7 according the reviewer’s suggestion.
Comments 12: Lines 391-393 and 399: It is difficult to understand how 100 mg leaf was weighed after homogenization with liquid nitrogen. In addition, 100 mg FW of leaf corresponds to a portion of a leaf, certainly not a whole leaf. It should be better explained and indicated which leaf was taken and at what age of the plant. “100 mg fresh leaf tissue” is too general.
Response 12: Thank you for pointing this out. We are very sorry for omitting the operational details of sampling in the method description. In the experiment, we first prepared the 1.5 mL centrifuge tubes with 1 mL lysis buffer. Then, we zeroed the centrifuge tube before we prepare to grind the sample. Thirdly, we used a spoon (it was kept in liquid nitrogen for pre-cooling) to quickly take the leaf powder and put in the tube for weighing, finally 100 mg leaf powder is gained and blended fully with lysis buffer. In addition, we taked about 3 cm long leaves including the inoculation wound for analysis at rice tillering stage. We have made supplements in the method 4.2 according to the reviewer’s suggestion.
Reviewer 2 Report
Comments and Suggestions for Authors
Dear Authors,
Congratulations for the nice work. It is well arranged, well written and experiments conducted neatly.
I have some little concerns which can be taken care of.
1. Statistics in Fig 4c.
2. In line 339 its 1000 bootstraps, in line 125 its 2000 bootstraps. Please select the proper one.
3. It would be good to show the autoactivation as a negative control on the AD and BD system.
4. How as these genes selected PR1a, PR1b, PR10? On what context? explain more explicitely.
5.Line 272 typo for START
6. OsAPX8 is also known to interact with with Os8N3/Xa13. can you please include a hypothetical model in the conclusion with your findings and other findings to propose a working model.
7. There are other APX in the plant system. Why only APX8 was found to be the partner?
8. How did you stumble on this protein "class III protein OsPCTP, which modulates the bacterial blight resistance of rice"? Mention more in the introduction and discussion. Why did you select SMART domains?
9. Just a general query, how does the membrane get effected by this gene in the transgenic plants? Is there a way for estimation membrane leakage/choline content etc.
Author Response
Comments 1: Statistics in Fig 4c.
Response 1: Thank you for pointing this out. We have made correction in figure 4c according the reviewer’s suggestion.
Comments 2: In line 339 its 1000 bootstraps, in line 125 its 2000 bootstraps. Please select the proper one.
Response 2: Thank you for pointing this out. We have made correction in manuscript.
Comments 3: It would be good to show the autoactivation as a negative control on the AD and BD system.
Response 3: Thank you for pointing this out. We have made the autoactivation as the negative control at same time. In figure 5A, the yeast co-expression combination BD/AD-APX8 and BD-PCTP/AD are two negative controls, which represent the autoactivation detection of two proteins respectively.
Comments 4: How as these genes selected PR1a, PR1b, PR10? On what context? explain more explicitely.
Response 4: Thank you for pointing this out. We have made supplements in manuscript. PR1a, PR1b and PR10 are the well-characterized PR genes in rice. The expression of PR1 genes is often used as a marker for plant systemic acquired resistance (SAR) development. PR10 encodes ribonucleases, which is considered to be able to degrade viral RNA during plant disease resistance process. In addition, PR1a, PR1b and PR10 are important SA signaling pathway reporter genes, which are induced by many pathogen infections. Moreover, we found a SA responsive element in the promoter region of OsPCTP. So, we selected these genes for analysis.
Comments 5: Line 272 typo for START
Response 5: Thank you for pointing this out. We have made correction in manuscript.
Comments 6: OsAPX8 is also known to interact with with Os8N3/Xa13. can you please include a hypothetical model in the conclusion with your findings and other findings to propose a working model.
Response 6: Thank you for comments and suggestions. As the reviewer pointed out, a hypothetical working model including PCTP-APX8 and Xa13 will help us to understand the role of PCTP in the resistance response to bacterial leaf blight and the hierarchical effects among PCTP, APX and Xa13. Unfortunately, On the one hand, our study here lacks sufficient genetic materials for PCTP and APX8, and their interactions have only been validated through molecular biology experiments. On the other hand, more molecular biology and genetic evidence is still needed to reveal the antagonistic relationship between PCTP and Xa13. Therefore, we suggest that the current evidence we made is insufficient for a working model to describe PCTP-APX8-Xa13. Based on this, we did not propose a working model in this article. But, We will conduct further research around this goal.
Comments 7: There are other APX in the plant system. Why only APX8 was found to be the partner?
Response 7: Thank you for pointing this out. As the reviewer pointed, there are a total of 8 APX members in rice. In this study, through yeast two hybrid screening library, we found that PCTP can interact with APX8 but not with other APX and the further BIFC and pull-down experiments also confirmed the interaction between PCTP and APX8. Therefore, this study only focuses on the role of his APX8-PCTP complex in disease resistance response.
Comments 8: How did you stumble on this protein "class III protein OsPCTP, which modulates the bacterial blight resistance of rice"? Mention more in the introduction and discussion. Why did you select SMART domains?
Response 8: Thank you for comments and suggestions. We have focused on the role of lipids (lipid metabolism and transport ) in plant growth, development, and stress response for a long time. The START domains are characterized as a hydrophobic ligand binding pocket and are involved in lipid/sterol binding, transport, and signaling in both animal and plant species. Then, most of START proteins (20 out of 26 proteins, class I proteins) in rice belong to HD-ZIP transcription factors because with HD-ZIP domains at same time. Previous studies showed that these transcription factors are generally associated with the growth and development of rice. In addition, orthologous genes EDR2 and WKS1 (belong to class II) were reported to modulate pathogen resistance in Arabidopsis and wheat, respectively. However, the functions of disease resistance for START domain proteins have not been reported yet. . So, in order to explore the new functions of START protein in rice, we chose OsPCTP which only contains START domain for research.
Comments 9: Just a general query, how does the membrane get effected by this gene in the transgenic plants? Is there a way for estimation membrane leakage/choline content etc.
Response 9: Thank you for your comments and suggestions. As the reviewer pointed that monitor the membrane situation change is the effective manner to reveal the function of rice PCTP. Unfortunately, we did not observe the changes in the membrane through microscopic methods. However, we have measured the changes in lipid composition in different genotype of OsPCTP materials (ospctp/ZH11/OsPCTP-OE) through metabolomics methods, and the results showed that the content of phosphatidylcholine in the over-expression material was significantly higher than that in the wild type (data not shown). Because the material used for this result is rice grains and it is not closely related to the role of PCTP in disease resistance response mentioned in this study, we did not present this data here.
Round 2
Reviewer 1 Report
Comments and Suggestions for Authors
Some really minor points need a revision:
Please remove the first paragraph of the abstract because the phosphatidylcholine transfer protein (PCTP) is neither the subject of the publication nor an outcome of the work, therefore, it is appropriate to introduce it but starting from the Introduction.
Line 81, ARABIDOPSIS should be added to “THALIANA MERISTEM LAYER 1 (ATML1).”
It should be indicated in the legend of Figure 1 that “Loc_Os02g26860” is a locus name not an “accession number.”
Comments on the Quality of English Language
The phrases added are written in approximate English and need to be slightly revised so that their meaning is well understood, e.g.:
“such as seedling and root at 14 d, leaf, leaf sheath, stem and inflorescence during the reproductive stage” (lines 154-155)
“The analytical tissues were about 3 cm long leaves including the inoculation wound at tillering stage” (lines 177-178)
“after 12 d of inoculation with Xoo inoculation at tillering stage” (line 200)
“The analyzed tissues were the leaves in the tillering stage” (lines201-202)
Author Response
Comment 1: Please remove the first paragraph of the abstract because the phosphatidylcholine transfer protein (PCTP) is neither the subject of the publication nor an outcome of the work, therefore, it is appropriate to introduce it but starting from the Introduction.
Response 1: Thank you for pointing this out. We have made correction according the reviewer’s suggestion.
Comment 2: Line 81, ARABIDOPSIS should be added to “THALIANA MERISTEM LAYER 1 (ATML1).”
Response 2: Thank you for pointing this out. We have made correction according the reviewer’s suggestion.
Comment 3: It should be indicated in the legend of Figure 1 that “Loc_Os02g26860” is a locus name not an “accession number.”
Response 3: Thank you for pointing this out. We have made correction according the reviewer’s suggestion.
Comment 4: The phrases added are written in approximate English and need to be slightly revised so that their meaning is well understood, e.g.:
“such as seedling and root at 14 d, leaf, leaf sheath, stem and inflorescence during the reproductive stage” (lines 154-155)
“The analytical tissues were about 3 cm long leaves including the inoculation wound at tillering stage” (lines 177-178)
“after 12 d of inoculation with Xoo inoculation at tillering stage” (line 200)
“The analyzed tissues were the leaves in the tillering stage” (lines201-202)
Response 4: Thank you for pointing this out. We have made corrections according the reviewer’s suggestion in the manuscript.